

# Improving machine learning detection of Alzheimer disease using enhanced manta ray gene selection of Alzheimer gene expression datasets

Zahraa Ahmed and Mesut Çevik

Department of Electrical and Computer Engineering, Altınbaş Üniversitesi, Istanbul, Turkey

## ABSTRACT

One of the most prominent neurodegenerative diseases globally is Alzheimer's disease (AD). The early diagnosis of AD is a challenging task due to complex pathophysiology caused by the presence and accumulation of neurofibrillary tangles and amyloid plaques. However, the late enriched understanding of the genetic underpinnings of AD has been made possible due to recent advancements in data mining analysis methods, machine learning, and microarray technologies. However, the "curse of dimensionality" caused by the high-dimensional microarray datasets impacts the accurate prediction of the disease due to issues of overfitting, bias, and high computational demands. To alleviate such an effect, this study proposes a gene selection approach based on the parameter-free and large-scale manta ray foraging optimization algorithm. Given the dimensional disparities and statistical relationship distributions of the six investigated datasets, in addition to four evaluated machine learning classifiers; the proposed Sign Random Mutation and Best Rank enhancements that substantially improved MRFO's exploration and exploitation contributed to efficient identification of relevant genes and to machine learning improved prediction accuracy.

## INTRODUCTION

Medical research has identified Alzheimer's disease (AD), also known as senile dementia, as the most common type of neurodegenerative disease that significantly impairs patient's capacity to perform daily activities. The development of intracellular neurofibrillary tangles due to tau hyperphosphorylation, the loss of neuronal tissue caused by gliosis proliferation, and the formation of extracellular amyloid plaques due to aberrant amyloid beta accumulation are among the pathological characteristic abnormalities of AD (*Hüttenrauch et al., 2018*). Reactive astrocyte morphology describes the molecular alterations that cause the brain cells to exhibit significant morphological changes in response to stressful conditions (*Preman et al., 2021*). In a healthy brain, astrocytes play several roles, such as promoting neuronal metabolism, maintaining the blood-brain barrier's integrity, and maintaining the equilibrium of ions in the extracellular space

Corresponding author
Zahraa Ahmed,
203720153@ogr.altinbas.edu.tr

(*Siracusa, Roberta & Salvatore, 2019*; *Vasile, Elena & Nathalie, 2017*). Despite its high prevalence, Alzheimer's disease is still one of most widely researched disease (*Holtzman, John & Alison, 2011*; *Rocca & Luigi, 2019*). In recent years, the field of molecular biology witnessed significant advancements due to integration of data mining and machine learning techniques in microarray analysis, and particularly in the diagnosis and treatment of molecular diseases. These computation approaches—such as rule mining, feature selection, clustering analysis, machine and deep learning—assist the identification of complex and intricate patterns in gene expression data. The application of these methods has not only improved the diagnosis accuracy, prognosis assessment, and patient's tailored-treatment but also significantly advanced our knowledge of the diseases at molecular level (*Tanveer et al., 2020*; *Shakir & Brittany, 2022*).

However, processing high-dimensional microarray datasets using machine learning algorithms is surrounded by several challenges, such as overfitting and computational complexity. The sparsity of high-dimensional data significantly reduces the prediction accuracy and other performance metrics of learning models (*Berisha et al., 2021*). Additionally, high dimensionality hinders the understanding of features impacts on models and reduces model interpretability, particularly in complex models like deep learning. To mitigate these issues, techniques for feature selection and dimensionality reduction are essential; as they retain the relevant information while reducing noise and redundancy, thereby enhancing prediction performance (*Remeseiro & Veronica, 2019*). Given the obstacles associated with the curse of dimensionality in AD prediction and early diagnosis, gene selection techniques are of great importance (*Osama, Hassan & Abdelmgeid, 2023*).

Despite the growing body of research on gene selection of Alzheimer disease using gene expression datasets, GSE. Several notable gaps and limitations persist in the reviewed literature. For instance, the traditional statistical filtering methods (*e.g.*, chi-square, mutual information, fold-change, and $p$-value), dimensionality reduction techniques (*e.g.*, principal component analysis (PCA)), and sequential iterative selection (*e.g.*, minimum-Redundancy maximum-Relevance (mRMR)) often lack the capacity to capture nonlinear and interactive gene relationships. The values—such as fold-change, $p$-value, and the threshold of cumulative variance sum in PCA—are arbitrarily defined and not sensitive to dataset-specifics, thereby can potentially exclude individual genes that are biologically and clinically relevant to the disease. Furthermore, the utilization of autoencoders transformation of input features into an abstract latent representation eliminates any connection between output and original features, as such, obscures the biological significance and analysis of relevant genes to the disease itself. While iterative sequential selection methods—such as mRmR—can be effective in small datasets, they are computationally intense and prone to suboptimal feature subsets in high-dimensional spaces.

Moreover, utilization of single swarm warper-based feature selection method—in this context, particle swarm optimization (PSO)—is inefficient to explore the huge feature space of GSE datasets. Traditional single swarm optimization algorithms are ill-suited for

large-scale optimization tasks, as they tend to prematurely converge, highly prone to local optima, and their performance is dependent on carefully tuned parameters.

Identifying gene interactions, hidden patterns, and effectively exploring the huge feature space in GSE Alzheimer datasets necessitate the use of sophisticated gene selection methods due to the intrinsic complexity arising from the large number of genes in the datasets. This study introduces a bio-inspired wrapper gene selection method based on the large-scale and parameter-free manta ray foraging optimization algorithm (MRFO). Accordingly, the aim is the potential reduction of Alzheimer's datasets through effective selection of relevant genes, thereby enhancing the predictive accuracy and aid in the early focused treatment of AD.

The specific objectives of this study are:

- To evaluate MRFO wrapper-based gene selection method.
- To investigate the appropriate MRFO enhancements with emphasis on efficient exploration of microarrays complex feature space and the selection of relevant genes aimed at enhancing prediction of Alzheimer disease.
- To assess the performance of enhanced gene selection method using four machine learning models.

For a guided reading, the rest of this article is arranged in sections, with "Related Work" providing a thorough overview and analysis of the current state of the literature on Alzheimer's disease gene selection techniques. "Materials and Methods" provides the background of microarray technology, data mining and machine learning approaches for Alzheimer's diagnosis, issues and challenges, and specifics of gene selection methods and techniques. The wrapper-based gene selection method is then explained, after which the manta ray foraging optimization algorithm, solution encoding, fitness function, and proposed enhancements are detailed. In "Enhanced Manta Ray Algorithm", the proposed algorithm is evaluated and the achieved results are thoroughly analyzed. "Enhanced Manta Ray Algorithm" reviews and compares the achieved results with reviewed literature. Last, conclusion and limitations of this study are summarized in "Conclusion".

## RELATED WORK

Numerous studies investigated gene selection of GSE microarrays to enhance diagnosis of Alzheimer's disease. Studies emphasizing feature selection strategies and meta-heuristic optimization algorithms will be the main focus of this literature review. Further, studies that utilize datasets other than the GSE datasets employed in this study are excluded from this review.

In consideration of the significant clinical applications, the identification of numerous regulatory relationships based on experimental studies is often expensive and sometimes unfeasible. A high-throughput co-expression network was proposed by *Zheng, Changgui & Huijie (2024)* in which genes are chosen according to their average path length and maximum connection to other genes. Through examination of gene expression data, the study estimated the cross-correlations between gene pairs and prioritized them to identify

the crucial value that splits the curve into two segments. As a result, modules are created inside the resulting network, and the representative biomarker is the gene with the highest degree and the shortest average path length. Applying this method to gene expression patterns in AD patients reveals that most of the suggested biomarkers coincide with those reported in the literature (*Zheng, Changgui & Huijie, 2024*). *Finney et al. (2023)* used various machine learning algorithms to do a meta-analysis of the cerebellum and frontal cortex of AD patients and healthy controls. Researchers contended that fold change, *p*-values, or a combination of the two are typically used in the conventional approach to identify genes that are differentially expressed. These methods have limits, though, and it is unclear whether they will provide enough data to make solid inferences regarding dysregulated genes that may be important for Alzheimer's disease (*Finney et al., 2023*). After applying principal component analysis (PCA) on more than 15,000 genes, the top 1,000 genes that showed the strongest correlation with the principal components were chosen for additional study. STRING v11 was used to do additional analysis on these genes to find gene-enrichment pathways and interaction networks. Thereafter, K-means clustering was applied to further improve the identification of biologically significant gene candidates by classifying genes into discrete network groups according to their connections (*Finney et al., 2023*).

*Ahmed & Suhad (2023)* suggested a wrapper-based Nomadic People Optimizer (NPO) for gene selection of Alzheimer's disease microarray datasets. The study shows that, in comparison to different selection methods, wrapper-based feature selection approaches are more successful at reducing the dimensionality of the data, preserving just those aspects that are directly related to the prediction model's classification accuracy. With an improved support vector machine (SVM) classification accuracy of 96% among other metrics, the proposed technique significantly enhanced the model's prediction of Alzheimer disease (*Ahmed & Suhad, 2023*).

The likelihood of developing of Alzheimer's disease in patients is significantly related to the activities of glucose metabolism related genes (GMRGs). According to *Wang et al.'s (2023)* study, abnormalities in glucose metabolism could act as early warning signs of AD. Researchers found that eighteen GMRGs—like glycolysis, the tricarboxylic acid cycle, and oxidative phosphorylation—to be strongly related to AD. The study identified 462 differentially expressed genes (DEGs) between AD and non-AD groups after DEGs visualization using heat and volcano maps. Next, using biological function and pathway analysis, weighted gene co-expression network analysis (WGCNA), gene ontology (GO), and Kyoto Encyclopedia of Genes and Genomes (KEGG) were used to further reduce the set of selected genes to just 12 genes. The selected genes were evaluated with AUC of 0.94 (*Wang et al., 2023*).

*Zhang et al. (2022)* investigated the coregulation of transcription factors, such as TBP and CDK7, and microRNAs by pathogenic genes linked to Alzheimer's disease. The study suggested multistep gene selection strategy that combines survival analysis, protein-to-protein interaction (PPI) network development, and differential expression analysis (*Zhang et al., 2022*). To increase the accuracy and usability of the data, the K-neighborhood technique was first employed to fill in the missing data. The selected

genes produced high area under the curve (AUC) values near 90% and revealed a connection to 15 drugs that specifically target these important genes; these drugs are mainly prescribed for a range of medical conditions, such as depression and pain management (*Zhang et al., 2022*). In another study, a framework that uses gene expression (GE) data to predict AD effectively was proposed by *El-Gawady, BenBella & Mohamed (2023)*. First, the study evaluates gene relevance using a variety of measures independently, including mutual information (MI), chi-squared ($\chi^2$), and analysis of variance (ANOVA). Several ML algorithms were selected such as support vector machine (SVM), random forest (RF), logistic regression (LR), and AdaBoost. Using a subset of 1,058 identified genes, the SVM model performed the best out of all the evaluated models, with an accuracy of 97.5% (*El-Gawady, BenBella & Mohamed, 2023*).

The study by *Mahendran et al. (2021)* aimed to use the gene selection hybrid technique based on gene expression data for efficient identification of relevant genes that frequently contain thousands of features but a limited number of samples. The main aspect of the suggested approach is the use of an improved deep belief network (IDBN) to improve the classification of Alzheimer's disease. The pipeline combines minimum redundancy and maximum relevance (mRmR), wrapper-based particle swarm optimization (WPSO), and autoencoders (AE). The suggested method outperformed conventional feature selection methods like PCA and correlation-based feature selection (CBFS) with a classification accuracy of 96.78% (*Mahendran et al., 2021*). *Chihyun, Jihwan & Sanghyun (2020)* investigated use of deep neural networks, gene expression, and DNA methylation profiles from diverse omics datasets to predict Alzheimer's disease. First, two-phase gene selection was applied to the heterogeneous omics datasets, which included differentially expressed genes (DEG) and differentially methylated positions (DMP) (*Chihyun, Jihwan & Sanghyun, 2020*). *Chihyun, Jihwan & Sanghyun (2020)* argued that conventional techniques for feature selection, such as least absolute shrinkage and selection operator (LASSO), Relief-F, and PCA can ensure the reduction of the number of features. However, biological relevance is not guaranteed by these methods (*Chihyun, Jihwan & Sanghyun, 2020*).

Biomarkers of Alzheimer's disease were investigated by *Long et al. (2016)* using two novel feature selection techniques based on support vector machines (SVM): support vector machine top forward selection (SVMTFS) and support vector machine forward selection (SVMFS). The best-performing proteins were iteratively selected using SVMFS. The process involved training an SVM model and evaluating its performance using leave-one-out cross-validation (LOOCV) accuracy, thereby optimizing the set of biomarkers for classification. On the other hand, SVMTFS chooses the next protein to be included from this ranking list in a preset order, ranking every protein according to the LOOCV accuracy of each SVM model. The diagnostic accuracy of both approaches was improved by ensuring the selection of biomarkers that significantly aid in differentiating Alzheimer's patients from healthy individuals (*Long et al., 2016*).

*Yao et al. (2019)* work focused on using computational predictions and experimental validation to identify blood biomarkers for Alzheimer's disease (AD). The study predicted possible blood-secretory proteins associated with AD using a classifier based on SVM and

trained on secretory protein data (*Yao et al., 2019*). First, two GSE datasets were used based on brain tissue from AD patients and healthy controls. The researchers employed the Benjamini and Hochberg approach to further adjust for false discovery rate (FDR), and considered probes with a *p*-value < 0.05. Following the mapping of the DEPs to their corresponding genes, 2,754 differentially expressed genes were identified, 296 of which were predicted to encode blood-secretory proteins essential for AD with AUC scores of 0.93 (*Yao et al., 2019*).

## MATERIALS AND METHODS

The roadmap of the conducted research is detailed in this section, beginning with an introduction to microarray technology and the use of machine learning and data mining in Alzheimer's disease diagnosis and the issues and challenges facing machine learning in Alzheimer's diagnosis and gene selection in biomedical applications. The section then further details data preprocessing, selection of machine learning algorithms, evaluation method, encoding techniques, and Manta Ray algorithm along with proposed enhancements: random sign mutation, best rank, and hybrid boundary function.

### Alzheimer disease and microarray technology

The field of genomics and molecular biology has been profoundly transformed because of the advancements in RNA and DNA microarray technologies. Microarrays, also known as DNA chips or biochips, are solid and compact platforms that are used to systematically immobilize DNA sequences. The process starts with hybridization, in which these immobile DNA sequences are allowed to interact with tagged RNA or DNA probes taken from a sample of interest; fluorescent dyes often serve as labels. The microarray is scanned to detect the fluorescent signals after removing any improperly attached probes. The amount of nucleic acid sequence in the sample is directly reflected in the fluorescence intensity at each location on the array, and this quantity correlates with expression level of the corresponding gene (*Bumgarner, 2013*; *Ehrenreich, 2006*; *Trevino, Francesco & Hugo, 2007*). Figure 1 illustrates the process of creating a microarray chip for a normal cell and another type of cancerous cell. From evaluation of gene expression profiles, researchers can identify co-expressed genes by comparing expression patterns, then, patterns are used to categorize biological samples based (*Trevino, Francesco & Hugo, 2007*; *Chatterjee & Nikhil, 2016*). Furthermore, the study of the regulatory processes underlying co-expression and the inference of genes' functional relationship can be initiated from co-expressed gene clusters. Gene expression patterns are then used as biomarkers in the context of tumor classification to categorize cancers and other subtypes of malignancies (*Van Dam et al., 2018*).

### Machine learning and data mining techniques in AD diagnosis

Data mining techniques and machine learning algorithms have become one of the most popular methods in microarray analysis. The integration of these methods has significantly transformed microarray analysis and advanced the detection and treatment of molecular diseases (*Swanson et al., 2023*). In the field of cancer diagnosis, for example, machine

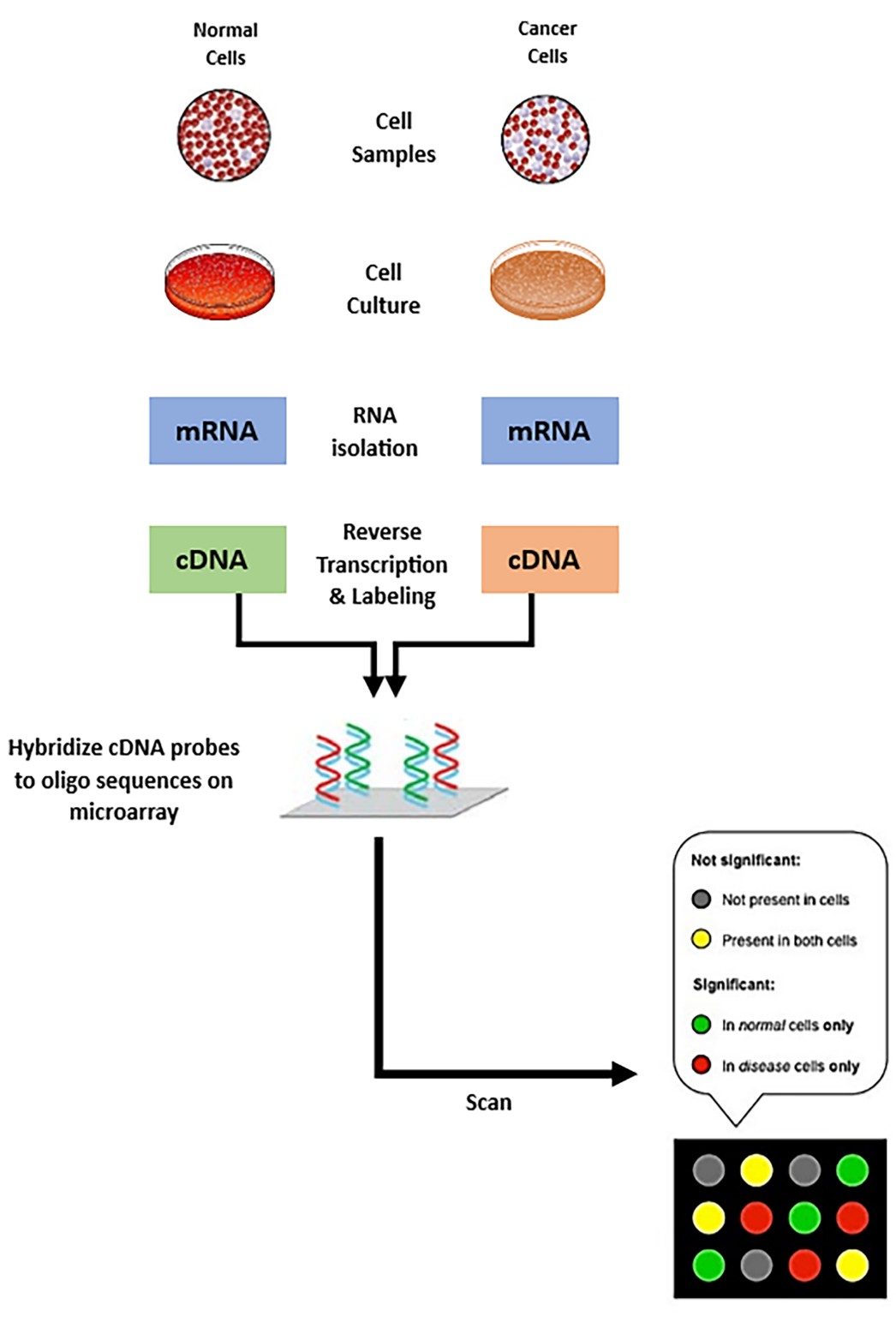

**Figure 1 Steps of creating microarray chips.**

learning models have the potential to discover oncogenic mutations and predict patient response to particular medicines, which allows for the development of individualized treatment programs. Importantly, large volumes of microarray data can be analyzed effectively using machine learning methods such as supervised learning, clustering techniques, and regression (*Pirooznia et al., 2008*). Compared to standard clinical approaches, machine learning algorithms can quickly and efficiently identify patterns and discover complex relationships among different gene expression levels, since each microarray measures the expression levels of hundreds of thousands of genes across multiple samples and many patients (*Pirooznia et al., 2008*). In the area of specialized medicine, this predictive ability is essential as treatment approaches are customized based on an individual's genetic profile (*Vadapalli et al., 2022*).

## Issues and challenges facing machine learning in AD diagnosis

Although supervised learning and data mining approaches are widely adopted in many fields, there are still many obstacles to overcome in microarray analysis using supervised learning and data mining approaches, among which is the "curse of dimensionality" (*Crespo, 2022*; *Verleysen & Damien, 2005*; *Bach, 2017*; *Altman & Martin, 2018*). This phenomenon pertains to a group of issues that emerge in the processing of high-dimensional datasets in machine learning and takes on particular significance when dealing with datasets pertaining to microarray genes. In simple terms, high-dimensional data become sparse with large feature dimensional space. This sparsity renders it difficult to detect data patterns using machine learning algorithms, thereby leading to inaccurate decisions. The risk of learning model overfitting is increased when there are more features or genes compared to small number of data samples. In addition, the computational complexity associated with processing a high number of dimensions as well as the delay in making medical decisions causing delays in disease diagnosis and treatment. This can result in poor classification performance of unseen test data samples due to weak learning generalization of training samples. In other words, not all genes in the microarray vast array of genes are relevant to the disease status. The inclusion of irrelevant features during model training subsequently impacts the classification performance of the disease by the learning model (*Yu, Yue & Michael, 2011*). Techniques for feature selection and dimensionality reduction are crucial to mitigate these challenges (*Zebari et al., 2020*; *Velliangiri & Alagumuthukrishnan, 2019*). On the one hand, feature selection ensures that only the genes from the vast feature space that are pertinent to the predictive model are retained. Consequently, increasing the model's interpretability, decreasing overfitting, and enhancing predictive accuracy. Furthermore, the computational complexity of microarray datasets can be exponentially decreased due to reduction of data dimensionality. This further contribute to less processing time particularly in sophisticated learning techniques like ensemble and deep learning. On the other hand, selection of relevant genes can help medical researchers in developing customized treatments that directly target certain genes and also improve treatment efficiency through personalized medicine by aiding in the customization of patients' care based on their unique genetic makeup. Last, the identification of gene biomarkers for any given disease can lead to a more specific and

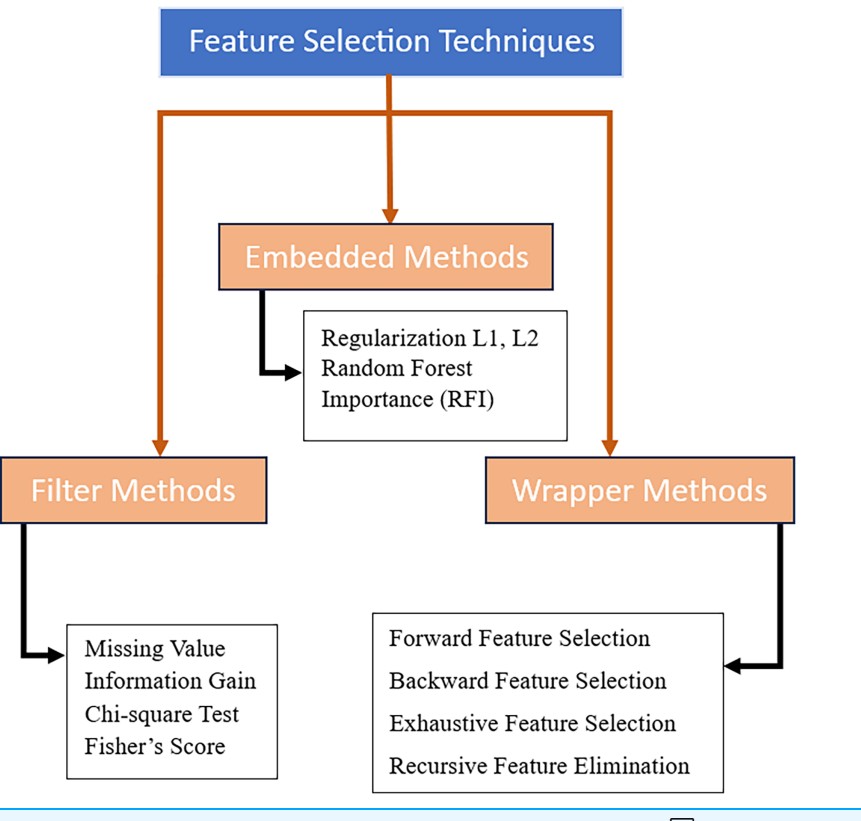

**Figure 2 Feature selection techniques.**

accurate diagnosis of the disease (*Van Cauwenberghe, Christine & Kristel, 2016*; *Loddo, Buttau & Di Ruberto, 2022*).

## Challenges of gene selection in biomedical applications

In the realm of machine learning, numerous techniques have been developed for efficient feature selection. These techniques, illustrated in Fig. 2, can be categorized into three primary groups: filter, wrapper, and embedded (*Chandrashekar & Ferat, 2014*; *Solorio-Fernández, Carrasco-Ochoa & Martínez-Trinidad, 2020*). The effectiveness of each strategy differs depending on the dataset and employed classifier. For example, chi-square and information gain are two filter-based techniques that are well-known for their computing efficiency and scalability to large datasets. Filter-based methods, on the other hand, handle features separately and disregard their interactions which leads to weak identification of intricate necessary genes and biomarkers for prediction or treatment development of the disease. Conversely, embedded techniques are considered to be more accurate than filter-based techniques as the feature selection is integrated into the algorithmic training process, allowing the learning model to internally evaluate different subsets of selected features iteratively (*Chandrashekar & Ferat, 2014*; *Solorio-Fernández, Carrasco-Ochoa & Martínez-Trinidad, 2020*).

In contrast, wrappers are a more advanced group of feature selection methods that consist of a strong search strategy to explore the feature space for potential features.

**Table 1 Alzheimer GSE datasets.**

| Dataset name | Dimension | |
|---|---|---|
| | Samples | Features |
| GSE1297 | 31 | 16,379 |
| GSE5281 | 161 | 54,674 |
| GSE33000 | 624 | 39,279 |
| GSE44768 | 329 | 39,279 |
| GSE44770 | 388 | 39,279 |
| GSE132903 | 195 | 42,178 |

Wrapper-based methods are independent of learning models and the selection of search strategies and learning models can be selected based on the specific case at hand (*Nogales & Marco, 2023*; *Dhal & Chandrashekhar, 2022*). One significant obstacle of using wrappers is the computational cost of searching through the vast search space of possible feature combinations. The iterative and sequential nature of feature elimination—such as exhaustive search, sequential forward and backward selection, and recursive feature elimination—are computationally not suitable for large microarray datasets due to their slow and weak capability to discover the optimal subset of features (*Chandrashekar & Ferat, 2014*; *Solorio-Fernández, Carrasco-Ochoa & Martínez-Trinidad, 2020*; *Dhal & Chandrashekhar, 2022*). Addressing this challenge requires the use of optimization algorithms as search methods for wrappers-based feature selection which also improves the search process. In contrast to the aforementioned strategies, the search process of large microarray datasets can be optimized through the exploration and exploitation of optimization algorithms like genetic algorithms (GAs), particle swarm optimization (PSO), and ant colony optimization (ACO) which require less time and computational resources. Additionally, the convergence to an optimal or near-optimal subset of features with improved generalizability and performance of the employed machine learning model is ensured by the strong and effective balance between exploration and exploitation of the optimization algorithm (*Dhal & Chandrashekhar, 2022*; *Kotsiantis, 2011*). In summary, optimization methods can effectively address issues and challenges related to search strategies in wrapper methods, especially in scenarios involving large microarray datasets with high-dimensional feature space (*Kotsiantis, 2011*; *Regan et al., 2019*; *Li et al., 2017*).

## Data preprocessing

Six of the most frequently used Alzheimer datasets in the literature have been selected to assess the optimization performance of the proposed gene selection algorithm. The demographics of the datasets are from 31 to 624 in number of samples and from 16,379 to 54,674 in number of features, detailed in Table 1. All datasets are obtained from the National Library of Medicine (NCBI) in comma-separated values (CSV) file format (*National Centre for Biotechnology Information, 2024*). For each dataset, the patients' meta-information, sample index and features details are removed, then data is organized in an $n \times m$ matrix, where $n$ is the number of patients and $m$ is the number of features. Last,

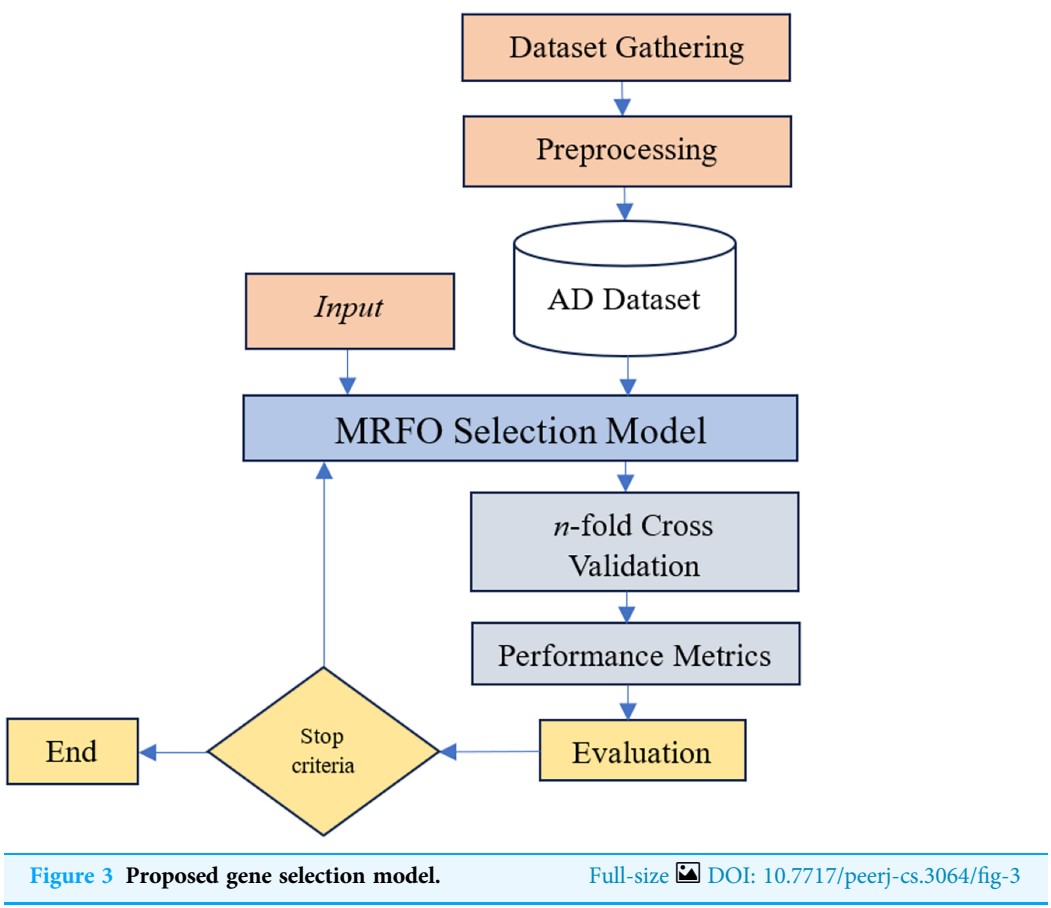

**Figure 3 Proposed gene selection model.**

label encoding is applied to generate the target or class label of each sample and the column is appended to the matrix, then converted and saved in CSV file.

## Experiment, machine learning algorithms, and configuration

Alongside the necessary libraries, the experiment was conducted using the PyCharm Community Edition integrated development environment (IDE) and Python version 3.9. The evaluation was performed on a computer equipped with 16 GB of system memory and a 2.4 GHz Intel Core i7 processor. To ensure consistency between the evaluation of the proposed approach and reviewed literature on Alzheimer's gene selection, four widely utilized classifiers—random forest (RF), XGBoost Classifier, naïve Bayes (NB), and SVM—were selected. A population size of 50 and a maximum of 100 optimization iterations were used in the assessment of the proposed gene selection method. For each evaluation, the training and testing are defined as 70% and 30%, respectively.

## Manta ray gene selection of Alzheimer's gene expression datasets

Given the difficulties caused by "curse of dimensionality" in predicting and diagnosing Alzheimer's disease, advance gene selection technique is necessary. The identification of intrinsic gene interactions, hidden patterns, and effective exploration of the large feature search space requires a sophisticated gene selection strategy due to the inherent complexity

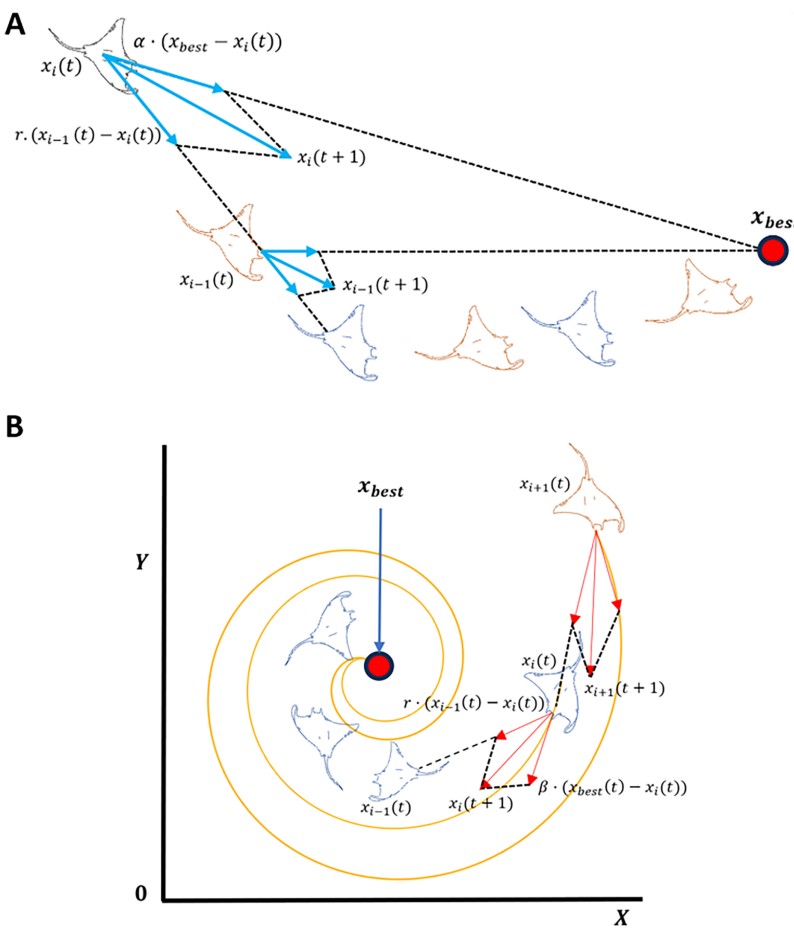

**Figure 4 Manta ray exploration and exploitation foraging techniques.** (A) Chain foraging, (B) Cyclone foraging.               

of the large number of genes in the GSE Alzheimer datasets. While wrapper-based gene selection methods are generally efficient in mitigating the impact of the "curse of dimensionality", their overall performance remains dependent upon the sensitivity of the underlying machine learning algorithm. MRFO is a parameter-free large-scale optimization algorithm potentially efficient to reduce the dimensionality of datasets and mitigate the effect of "curse of dimensionality" in Alzheimer's gene expression dataset. The proposed approach in this study has the following key steps illustrated in Fig. 3.

## Manta ray foraging optimization

MRFO is a metaheuristic optimization algorithm inspired by the foraging behavior of manta rays (*Zhao, Zhenxing & Liying, 2020*). The algorithm is comprised of three main foraging strategies: chain, cyclone, and somersault. Chain focuses on local search, while cyclone foraging prioritizes global exploration as shown in Fig. 4. The algorithm is parameter-free and computationally efficient, making it well-suited for large-scale optimization tasks (*Zhao, Zhenxing & Liying, 2020*). In chain foraging, manta rays establish a feeding chain when they arrange themselves head-to-tail; each one, except for

the first, moves towards the manta ray in front of it. In other words, each manta ray adjusts its position in relation to the entity immediately pre-ceding it as well as the current global optimal solution. This chain foraging model can be expressed mathematically as follow (*Zhao, Zhenxing & Liying, 2020*):

$$x_i^d(t+1) = \begin{cases} x_i^d(t) + r.\left(x_{best}^d(t) - x_i^d(t)\right) + \alpha.\left(x_{best}^d(t) - x_i^d(t)\right) & i = 1 \\ x_i^d(t) + r.\left(x_{i-1}^d(t) - x_i^d(t)\right) + \alpha.\left(x_{best}^d(t) - x_i^d(t)\right) & i = 2, \ldots, N \end{cases} \quad (1)$$

$$\alpha = 2.r.\sqrt{|\log(r)|} \quad (2)$$

where $x_i^d(t)$ is the position of the $i$th individual at time t in the $d$th dimension. $r$ is a random vector within the range of [0, 1], and $\alpha$ is a weight coefficient. $x_{best}^d(t)$ denotes the plankton with the highest concentration. Cyclone foraging is the second foraging strategy. Manta rays congregate when there is a high concentration of plankton, leading to spiral formations due the alignment of tails and heads (*Zhao, Zhenxing & Liying, 2020*). This behavior enhances global exploration by moving in spiral paths toward the best solution while following the ray ahead. To simplify, cyclone foraging can be expressed mathematically as:

$$x_d^i(t+1) = \begin{cases} x_{best}^d + r.\left(x_{best}^d(t) - x_i^d(t)\right) + \beta.\left(x_{best}^d(t) - x_i^d(t)\right) & i = 1 \\ x_{best}^d + r.\left(x_{i-1}^d(t) - x_i^d(t)\right) + \beta.\left(x_{best}^d(t) - x_i^d(t)\right) & i = 2, \ldots, N \end{cases} \quad (3)$$

$$\beta = 2e^{r1\frac{T-t+1}{T}}.\sin(2\pi r_1) \quad (4)$$

where $\beta$ is the weight coefficient, T is the maximum number of iterations, and r1 is a random number within the range of [0, 1]. Last, being a random, regular, local, and cyclical movement, somersault helps manta rays increase their food intake; the food is considered as a pivot around point enabling Mantas to flip and attain new positions (*Mirjalili & Andrew, 2016*). This behavior can be modeled mathematically as:

$$x_i^d(t+1) = x_i^d(t) + S.\left(r_2.x_{best}^d - r_3.x_i^d(t)\right), \quad i = 1, \ldots, N \quad (5)$$

where $S$ is the somersault factor that determines the range of somersaults performed by manta rays, with $S = 2$, $r_2$ and $r_3$ are two random numbers within the range of [0, 1].

## Solution encoding

Generally, for a given optimization task, the solution vector $\vec{\pi}$ is composed of $N$ continuous values, where $N$ is equal to the *Dimension* of the problem. Each $\vec{\pi}$ represents a unique solution within the population of solutions. The fitness function is used to evaluate each individual solution while the optimization process explores and exploits the search space and iteratively updates the population of solutions. In gene selection optimization, each value in the solution vector $\vec{\pi}$ must indicate the "inclusion" or "exclusion" of the specific gene from the dataset. Therefore, continuous values are transformed before the evaluation step. During the transformation process, one of binary encoding functions are used and continuous input values are transformed to a binary string of 0s and 1s, where 0 indicates gene/feature exclusion while 1 indicates specific gene/feature inclusion. There are two types of transformation functions, S-Shaped and V-Shape, each of which consists of a

Table 2 S-shaped and V-shaped transformation functions.

| Type | Name | Function |
|---|---|---|
| S-shaped | S1 | $S(x) = (1/(1 + e^{-2x})$ |
| | S2 | $S(x) = 1 / (1 + e^{-x})$ |
| | S3 | $S(x) = 1 / (1 + e^{\left(\frac{-x}{2}\right)})$ |
| | S4 | $S(x) = 1 / \left(1 + e^{\frac{-x}{3}}\right)$ |
| V-shaped | V1 | $V(x) = \mid \mathrm{erf}(\,(\sqrt{x}\,/\,2)\, * x\,)\mid$ |
| | V2 | $V(x) = \mid \tanh(x)\mid$ |
| | V3 | $V(x) = \mid x\,/\,\sqrt{1 + x^2}\mid$ |
| | V4 | $V(x) = \left\mid \dfrac{2}{\pi}\,arc\tan\left(\dfrac{2}{\pi}x\right)\right\mid$ |

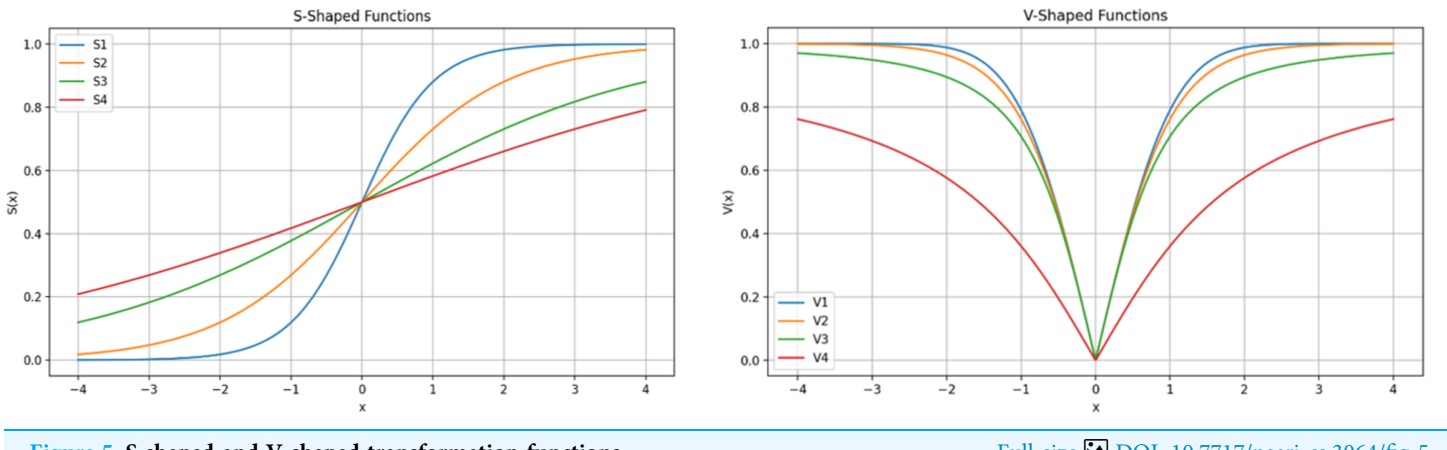

Figure 5 S-shaped and V-shaped transformation functions.

family of four functions, as shown in Table 2 and in Fig. 5 (*Ghosh et al., 2021*; *Mirjalili & Andrew, 2013*). The specific details of the optimization problem at hand and the features of the solution determine the choice of the specific function. When sensitivity to input changes is required, the S-shaped transformation functions provide a progressive and seamless transition from status 1 to 0 and vice versa. Whereas V-Shaped provides a very sharp transformation decision that is required in situations where input changes require a decisive response.

S-shaped transformation function S1 was selected in this work to encode the solution $\vec{\pi}$ values into the binary form. The steps of the solution encoding/evaluation can be summarized as follow:

1. The continuous values for every solution vector $\vec{\pi}$ in the population are encoded using the S1 function before evaluation.
2. Then, the active set of genes is determined using the binary-encoded vector $\vec{\pi}$ based on the index of the 1s values. New dataset $\bar{D}$ is then created with $N \times \bar{M}$ dimension, where

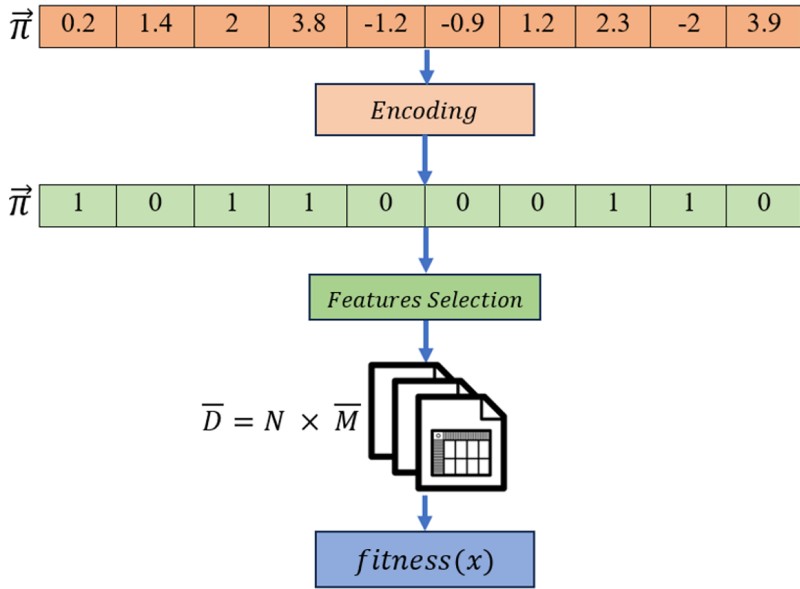

**Figure 6 Solution encoding steps in the proposed algorithm.**

$N$ denotes the number of samples and $\bar{M}$ is the number of selected features, satisfying $\bar{M} \leq M$; $M$ is the original number of features.

3. The process of encoding the continuous values to binary is done using the following:

$$Encode(x_i) = \begin{cases} 1 & if\ rand\ \leq S1(x_i) \\ 0 & else \end{cases} \tag{6}$$

where *rand* is a random number ranged between [0,1]. When the value of a randomly generated number is less than or equal to $S1(x_i)$, an encoded value of 1 is assigned, meaning the selection of the gene at *ith* index. A complete example of solution encoding in this study is visualized in Fig. 6.

4. Then, five-fold stratified cross-validation is used to divide the newly formed $\bar{D}$ into five-folds of training and testing. Stratified cross-validation ensures the reduction of variance and bias especially in training and testing of imbalanced datasets.

5. Classification performance is then evaluated.

6. Optimization fitness function is applied to the obtained performance metrics and a fitness value is assigned to determine the effectiveness of the solution vector $\vec{\pi}_i$.

## Evaluation metrics and fitness function

The nature of the optimization problem determines the set of performance metrics to be evaluated. In this study, the classification accuracy of the learning algorithm after gene selection is determined based on the mean accuracy of five-fold stratified cross-validation. In the context of optimization, objectives or performance metrics are often combined into a single score *via* optimization fitness function. Such that, the score is utilized to compare

two solutions quantitatively based on how "*fit*" one solution is compared to another. Given the nature of comparable or contradictory objectives, the weighted sum approach can successfully navigate the trade-offs among several desirable performance metrics. Weights are assigned to prioritize metrics in the objective function. Based on several reviewed feature selection fitness functions (*Brezočnik, Iztok & Vili, 2018*; *Naik, Venkatanareshbabu & Damodar, 2020*; *Cinar, 2023*), the formulated fitness function used in the proposed gene selection approach is as follows:

$$fitness = (1 - accuracy_{mean}) + w * \frac{SF}{\forall F} \tag{7}$$

where $accuracy_{mean}$ is the mean accuracy of five-fold stratified cross-validation, $w$ is features weight, $SF$ is the number of selected features, and $\forall F$ is the number of features in the dataset.

## ENHANCED MANTA RAY ALGORITHM

MRFO algorithm (Algorithm 1) was extensively evaluated using the selected Gene Expression Omnibus datasets to preliminary assess its gene selection performance. This preliminary evaluation step is a crucial analysis step for the assessment of MRFO performance in relation to both small and large feature spaces. The number of optimization iterations were systematically altered (100, 300, and 500), different encoding functions were utilized, and the outcomes were thoroughly examined. Despite the relatively good classification accuracy among four evaluated classifiers, MRFO algorithm demonstrated significant weaknesses in terms of the number of selected genes and scalability efficiency. MRFO consistently attained high number of genes, averaging from 30% to—occasionally exceeding—45% of the full dataset feature set. For instance, MRFO averaged—in ten independent runs—between 7,135 to 7,433 features out of 16,379 for GSE1297 and nearly half of the total features for large GSE5281 dataset, 24,462 out of 54,674. A similar behavior was also observed for mid-size GSE44768, GSE44770 and GSE33000 datasets (39,279 features), MRFO averaged feature selection between 17,978 to 18,222, 18,211 to 19,078, and 18,776 to 19,063, respectively. This suggests that MRFO lacks effective and strong exploration-exploitation required for dimensionality reduction of GSE datasets. Moreover, the algorithm's inability to significantly reduce feature sets even when optimization iterations was increased from 100 to 300 and 500 further highlights MRFO weak exploration-exploitation of solution search space. Increasing the number of optimization iterations—particularly in medical applications—is discouraged and often avoided, as it leads to prolonged decision-making process undesired in time-sensitive medical scenarios. It should also be noted that experimenting with different encoding functions (S1 to S4) had negligible to marginal effect on MRFO selection behavior performance, which further emphasize the irrelevance of explored solutions to the classification of Alzheimer disease across four evaluated classifiers.

In light of the noted drawbacks and limitations, this study suggests three key improvements to MRFO with focus on improving convergence and feature space

---

**Algorithm 1** Manta ray foraging optimizer (MRFO).

1    **Input:** Population size $N$, maximum number of iterations $T$

2    **Output:** $x_{best}$: set of best selected genes

3    **Procedure:**

4    Define fitness function $f(x)$: Eq. (7)

5    **For** every individual solution $Ind_i$ in population $POP$, $i = 1 \ldots \ldots N$, $t = 1$:
     Initialize each manta ray such that: $x_i = LB + rand(UB - LB)$, where
$i = 1 \ldots \ldots Dim$,
        $LB$ is the lower bound, $UB$ is the upper bound, and $Dim$ is the problem space.
     Apply solution encoding: Eq. (6)
        Evaluate objective function $f(x)$, such that $fit_i(t) = f(Ind_i(t))$

6    Obtain the best $fit_{best}$ solution: $x_{best}$.

7    **FOR** $It = 1$ to $T$ **Do**

8        **IF** rand < 0.5 **Then**

9            Perform Cyclone Foraging from Section 4.2.1 and generate $POP_{new}$

10       **Else:**
             Perform Chain Foraging from Section 4.1.1 and generate $POP_{new}$
                 For every $Ind_i$ in $POP_{new}$, $i = 2 \ldots \ldots N$
                     Apply **Sign Random Mutation (SRM)**

12       **END IF**

13       **For** every individual solution $Ind_i$ in population $POP_{new}$, $i = 1 \ldots \ldots N$:
             **Apply Hybrid Boundary Function (HBF)**
             Apply solution encoding: Eq. (6)
             Evaluate objective function $f(x)$, such that $fit_i(t + 1) = f(Ind_i(t + 1))$
             IF $fit_i(t + 1) < fit_i(t)$, then $Ind(t) = Ind(t + 1)$
         **END FOR**

14       Perform Somersault Foraging on generated $POP_{new}$ from Section 4.1.3

15       **For** every individual solution $Ind_i$ in population $POP_{new}$, $i = 1 \ldots \ldots N$:
             **Apply Hybrid Boundary Function (HBF)**
             Apply solution encoding: Eq. (6)
             Evaluate objective function $f(x)$, such that $fit_i(t + 1) = f(Ind_i(t + 1))$
             IF $fit_i(t + 1) < fit_i(t)$, then $Ind(t) = Ind(t + 1)$ and
$fit_i(t) = fit_i(t + 1)$
                 IF $fit_i(t + 1) < fit_{best}$, then $x_{best} = Ind_i(t + 1)$ and $fit_{best} = fit_i(t + 1)$

15       **END FOR**

16       Apply **Best Rank (BR)** on $POP_{new}$

17   **END FOR**

18   **Return** $x_{best}$

---

exploration to enhance Alzheimer's disease prediction accuracy. The proposed improvements to MRFO in this study are highlighted in the following pseudocode (Algorithm 1) and detailed in the subsequent sections.

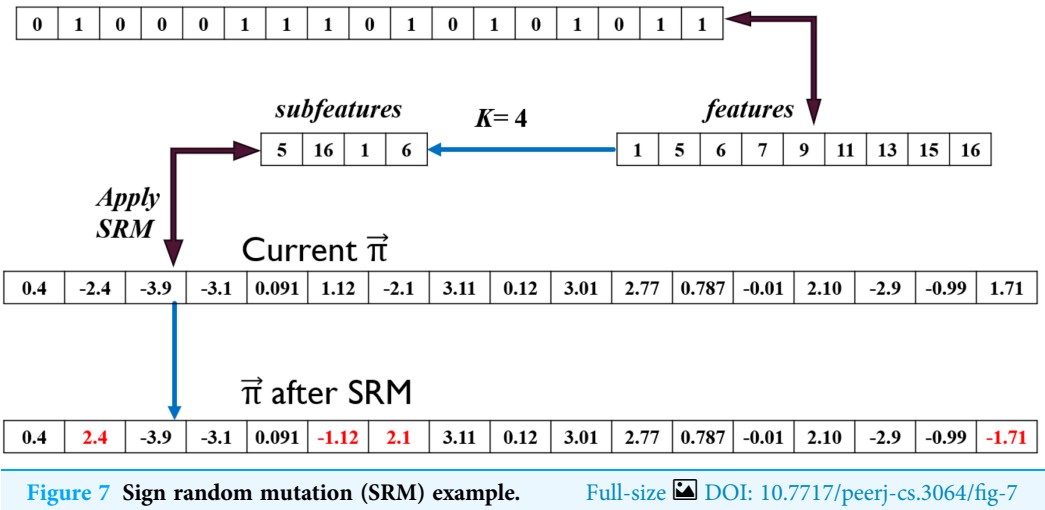

**Figure 7** Sign random mutation (SRM) example.

## Enhancing MRFO exploration through best rank

Manta rays engage in a behavior known as chain foraging in which they swim towards allocates plankton, as shown in Fig. 4A. The concentration of plankton at a given position determines the attractiveness of that position. Population-based optimization methods are hierarchical, whereby the population's top solution vector, $\vec{\pi}_0$, exerts the greatest influence over the population. Such influence progressively diminishes for the subsequent solutions, such as the second, third, and so forth. The position of each solution $\vec{\pi}$ is updated during each iteration based on the current best solution and the solution ahead of it, such that the solution of $\vec{\pi}_i(t)$ is affected by the solution position $\vec{\pi}_{i-1}(t)$.

The Solution Rank approach is a process of ranking solutions based on their estimated fitness values from best to worst. The first proposed enhancement of MRFO in this work is to improve exploration. The selection of multiple best solutions or pool of elite solutions can increase the time complexity of an optimization algorithm which is critical especially within the context of Alzheimer high-dimensional datasets (*Tang et al., 2021*). The computational cost is greatly increased by the expansion of the combinatorial space of gene subsets, thereby reducing the viability of implementing the solution. Therefore, the proposed approach suggests only replacing the population's first solution, or population *Head*, with the population's highest-ranked solution, or $local_{best}$, at each optimization iteration such that $Local_{best} \neq Global_{best}$ and $fitness$ of $Local_{best} < Population_{Head}$. With $Local_{best} = Global_{best}$, the second highest ranked solution is then used. Swapping $Local_{best}$ guarantees that the remaining population is guided to better search area at every iteration and improves exploration while ensuring a minimal increase in time complexity.

## Enhancing MRFO exploitation through sign random mutation

Pertaining to the MRFO cyclone foraging phase shown in Fig. 4B, a group of manta rays forms long spiral foraging chain when mantas come across a cluster of planktons such that every member of the swarm moves closer to the food source in a spiral pattern. This strategy greatly improves exploitation as well as exploration. From rigorous evaluation of

MRFO, varying transformation function from S-Shaped to V-Shaped (refer to Table 2, S1 and V1) considerably decreased the number of selected genes while maintaining low prediction accuracy measured at less than 60%. This drop in accuracy suggests that rather than the quality of the selected genes, the selection was driven by the sharp definitive cutoff transition of V1 V-shaped function. In addition, weak genes correlations and different patterns in AD GSE datasets further contributed to the poor classification accuracy. Based on the previous observations, a novel sign random mutation (SRM) is proposed to enhance MRFO exploitation. The three main steps of the proposed method are as follows:

1. All active (refer to Fig. 7, *i.e.*, has the value of 1) genes indexes from $Global_{best}$ solution are grouped into an array called *Activefeatures*.
2. For a length of $K$, a set called *Subfeatures* of randomly selected indexes are chosen from *Activefeatures*. Value of $K$ is calculated as follow:

$$K = int\left( \frac{length(Activefeatures)}{1 + r} \right) \tag{8}$$

$r$ is a random number generated between $[0, 1]$.

Then, all the values in the solution vector $\vec{\pi}$ are masked and the sign of the values indexed in *Subfeatures* are changed by applying the following:

*for each* **gene index** $e$ *in Subfeatures* list:
$\vec{\pi}[e] = \vec{\pi}[e]^* - 1.$

Guided by active genes in $Global_{best}$, the values in solution $\vec{\pi}$ indexed at $[5, 16, 1, 6]$ are changed from $[1.12, 1.71, -2.4, -2.1]$ to $[-1.12, -1.71, 2.4, 2.1]$ as shown in Fig. 7. This is translated to an increase or decrease in gene activation probability. For instance, the probability of generating a random value $r$ between $[0, 1]$ such that $r \leq S(-2.4)$ is significantly lower than that when $r \leq S(2.4)$, where $S$ is the S1 transformation function. Increasing probability raises the likelihood of the second gene being activated in the newly derived solution. Conversely, the activation probability was much higher for the *6th* gene before applying SRM which implies higher likelihood of this gene being deactivated.

## Hybrid boundary function

The values in the solution vector $\vec{\pi}$ generated by the optimization algorithm must always be within predefined boundaries; the search space limits are set accordingly to ensure solution validity. All values $v$ in $\vec{\pi}$ must satisfy $LowerBound \leq v \leq UpperBound$ and exceeding these limits means that the value is must be either clamped to the nearest bound or replaced with a new random value (*Xu & Yahya, 2007*; *Huang & Ananda, 2005*). In both cases, the algorithm's convergence is negatively influenced. First, based on the S-shaped and V-shaped transformation methods, the new clamped value implies the feature is more likely to be permanently active or inactive. Referring to Fig. 5, $\forall values < 4$ are always clamped to 4, and since the probability of $Encode(S1(4))$ (see Eq. (6)) is more likely to be 1, then, the set of all clamped values will consequently always be active. This assertion is also valid for $\forall values < -4$, such that, all clamped values will be excluded. It can be argued that

---

**Algorithm 2** Hybrid boundary function.

1  **Input**: solution vector $\vec{\pi}$
2  **Procedure**:
3      **If** coefficient < random **r**: # *Clamping*
4          **For** each value $v_i$ in $\vec{\pi}$:
5              if $v_i$ is out of boundary: $v_i$ = nearest bound,
                  such that if $v_i \leq LowerBound$, then $v_i = LowerBound$.
                  and if $v_i \geq UpperBound$, $v_i = UpperBound$.
6          **End For**
7      **Else**: # *Crossover*
8          **For** each value $v_i$ in $\vec{\pi}$:
9              if $v_i$ is out of boundary:  $v_i = v_i^{Global_{best}}$
10         **End For**
11     **Return** $\vec{\pi}_{new}$

---

the impact of the clamped approach decreases as the optimization progresses. However, the validity of this assumption depends on several factors, such as the position of the optimal solution in the search space, the current search direction of optimization algorithm, and the algorithm's strong balance of exploration and exploitation.

Second, the reinitialization of value $v_i$ in $\vec{\pi}$ to a completely new value may lead to an interruption in search continuity caused by a change in the search direction towards the optimal solution. And to greater extent, reducing exploration efficiency by revisiting the previously explored solutions. As a result, a hybrid boundary function (HBF) (Algorithm 2) is proposed in this work based on clamped and improved reinitializations using genetic crossover between $v_i$ in $\vec{\pi}$ and $v_i$ in $Global_{best}$ to mitigate the effect of search interruption. At early iterations of the optimization process, the first method is primarily utilized given that coefficient < $r$, where $r$ is a random number between [0, 1]. When the value of optimization coefficient then increases in proportion to the optimization iteration number $iT$, the improved reinitialization is more likely to be utilized. This selection was necessary due to the dependency of the proposed improved reinitialization approach on $Global_{best}$. The optimization coefficient is calculated as follows:

$$Coefficient = \frac{Current\ Iteration}{Max\ Iteration}. \tag{9}$$

It should be emphasized that several variations of HBF were tested. For example, testing *Sign Clamping* did not increase MRFO exploration and exploitation efficiency. The performance declined as the optimization progresses and the observed changes in the number of active genes were chaotic. Similarly, sign inverse reinitialization between $v_i$ in $\vec{\pi}$ and $v_i$ in $Global_{best}$ achieved marginal improvements during the early stages of optimization. Further still, the algorithm didn't efficiently reduce the number of genes as well as failed to improve classification accuracy. This behavior was the result of the contradictory behavior of proposed boundary function and sign random mutation, which

**Table 3** **EMRFOGS performance and metrics measurements on the GSE1297 dataset with 16,379 features.** The bold and underlined entries in this table indicate the best value.

| Model | Measurements Before/After | | | | AUC scores (after) | Selected features | $Global_{best}$ convergence iteration |
|---|---|---|---|---|---|---|---|
| | Accuracy | F1 | Recall | Precision | | | |
| NB | 0.60/**1.0** | 0.60/**1.0** | 0.60/**1.0** | 0.60/**1.0** | **1.00** | 24 | 95 |
| RF | 0.70/0.80 | 0.57/0.76 | 0.70/0.80 | 0.48/0.84 | 0.90 | **4** | 95 |
| SVM | 0.70/0.80 | 0.57/0.76 | 0.70/0.80 | 0.48/0.84 | 0.52 | 5 | **83** |
| XGBoost | **0.70/1.0** | **0.67/1.0** | **0.70/1.0** | **0.67/1.0** | **1.00** | 6 | 91 |

caused chaotic search interruptions and subsequently decreased the efficiency of the generated solutions. The pseudocode for the HBF function is shown below.

## RESULTS

### GSE1297

The performance metrics of four machine learning models—RF, SVM, eXtreme Gradient Boosting (XGBoost), NB—before and after applying EMRFOGS gene selection are depicted in Table 3. NB shows uniform performance across all performance metrics. Comparing RF and SVM classifiers to NB and XGBoost classifiers, both performed identically and had lower precision scores which suggests poor categorization with a high rate of false positives. With the highest F1-score of 0.67 and highest scores in accuracy, recall, and precision, the XGBoost classifier showed the best performance as it maintained strong classification with an efficient balance between recall and precision.

After applying enhanced manta ray foraging optimizer gene selection (EMRFOGS), NB, and XGBoost both attained perfect scores of 1.00 for each performance metric, including AUC. Such behavior can be interpreted as either potential of highly effective gene selection or possible overfitting of the classifier. Effective gene selection in this instance means that the selected subset of genes offered thorough representation of the underlying relationships of data that were essential for classifier prediction. EMRFOGS-RF and EMRFOGS-SVM showed significant gain across all metrics, with increases of ~14%, 33%, 14%, and 75% in classifier accuracy, F1, recall, and precision, respectively, as detailed in Table 3.

With efficient and balanced exploration and exploitation, EMRFOGS-RF reduced false positives and negatives and enhanced the selection of relevant features throughout optimization iterations and attained the second-highest AUC scores of 0.90. This can be seen in Fig. 8 where the minimum number of selected features begins to converge from early iterations, around iteration 20. The initial variability in the maximum number of features indicates that EMRFOGS-RF actively exploring different solution dimension (number of genes) then progressed gradually to exploitation. The improvement in performance metrics across four distinct machine learning models generally shows the effect of the suggested EMRFOGS gene selection on GSE1297; it illustrates the efficiency of the suggested gene selection strategy in AD prediction. The models responded differently in terms of improvements in performance measures. Therefore, it is not feasible to assess

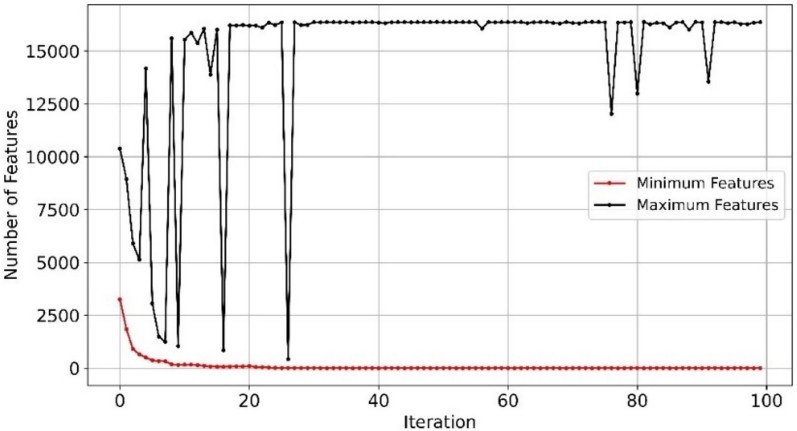

**Figure 8** Minimum and maximum number of features selected by EMRFOGS from GSE1297 using RF classifier.

**Table 4 EMRFOGS performance and metrics measurements on the GSE5281 dataset with 54,674 features.** The bold and underlined entries in this table indicate the best value.

| Model | Measurements before/after | | | | AUC scores (after) | Selected features | *Global_best* convergence iteration |
|---|---|---|---|---|---|---|---|
| | Accuracy | F1 | Recall | Precision | | | |
| NB | 0.85/0.87 | 0.85/0.87 | 0.85/0.87 | 0.88/0.88 | 0.92 | **162** | **70** |
| RF | 0.89/0.91 | 0.89/0.91 | 0.89/0.91 | 0.90/0.91 | **0.99** | 238 | 72 |
| SVM | **0.93/0.95** | **0.93/0.95** | **0.93/0.95** | **0.94/0.96** | 0.98 | 165 | 89 |
| XGBoost | 0.85/0.91 | 0.85/0.91 | 0.85/0.91 | 0.87/0.91 | 0.98 | 220 | 89 |

the generalized performance due to the possibility of overfitting in the NB and XGBoost classifiers. However, RF can be considered the best classifier with only four selected genes.

## GSE5281

From the obtained results depicted in Table 4, XGBoost and naïve Bayes performed equally in all of classification metrics—similar to GSE1297 dataset—with NB achieving higher precision scores of 0.88 compared to 0.87. With respect to classification accuracy, SVM achieved superior performance attaining the highest accuracy, F1, and recall scores of 0.93, and 0.94 in precision which further highlights its prediction competence of AD GSE datasets. In contrast to GSE1297, RF outperformed both NB and XGBoost classifiers in all metrics. SVM remained the top performing classifier with the highest scores among all evaluated metrics after EMRFOGS gene selection—see Table 4—scoring 0.95 for all performance metrics and 0.96 in precision. The classification results of RF and XGBoost classifiers were similar, with RF attaining the highest AUC scores of 0.99. In addition, EMRFODS gene selection significantly enhanced XGBoost prediction accuracy by ~7% in comparison to RF. This indicates an improved classification sensitivity following gene reduction. Although, NB was the least-performing classifier, it attained the lowest number of genes—only 162 out of 54,674—in comparison to the other three evaluated classifier.

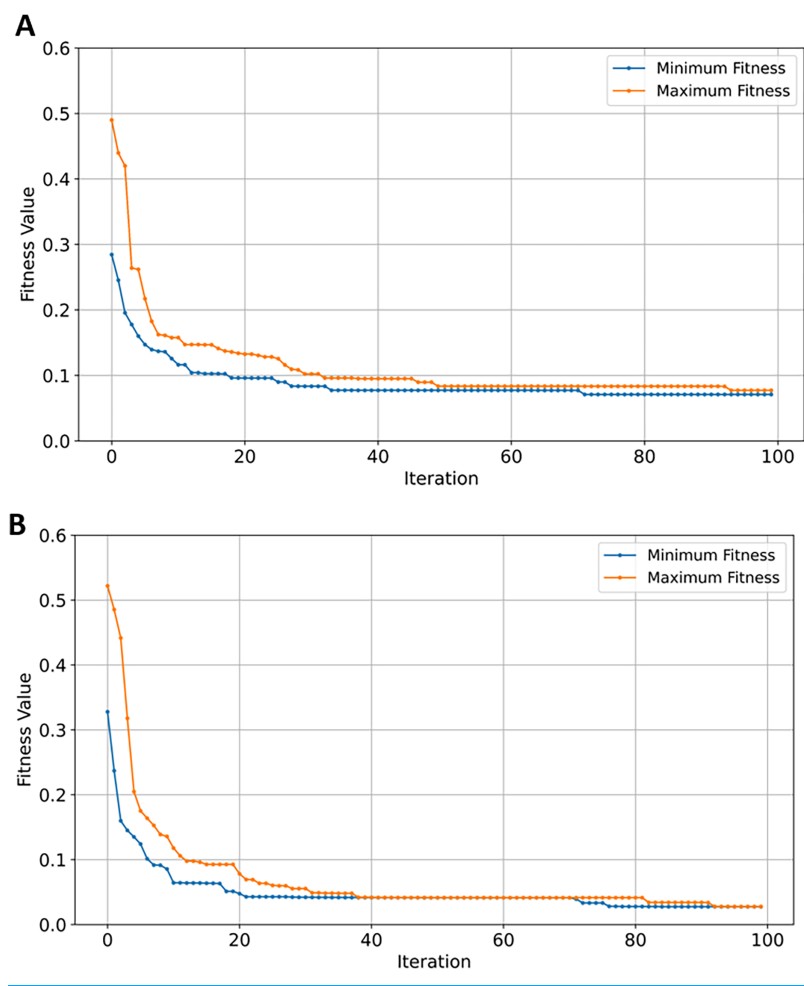

**Figure 9 Convergence of fitness minimum and maximum values of MERFOGS for GSE5281 dataset.**
(A) Naive Bayes classifier, (B) SVM classifier.

This suggests that NB was not sensitive enough and converged near iteration 70 with no further improvement in prediction accuracy, as shown in Fig. 9A. In comparison to SVM, the classifier efficiently converged at iteration ~89 with 165 relevant genes to AD prediction, see Fig. 9B. Overall, EMRFOGS performance in GSE5281 shows a strong balance between exploration and exploitation, the algorithm effectively navigated the complex feature-rich search space regardless of the utilized classifier.

### GSE33000

The XGBoost classifier outperformed the rest of the classifiers in the initial evaluation before gene selection, with an accuracy, F1, recall, and precision of 0.92 (see Table 5). RF performed well as the second-best classifier with roughly 2.22% lower scores in all metrics in comparison to XGBoost. SVM performance is approximately 7.19% worse than that of RF (see Table 5) while NB was the worst-performing classifier. Further still, NB scores are ~29.57% worse than best-performing classifier in comparison to only ~9.41% when tested on the GSE5281 dataset. The inefficient performance can be attributed to classification bias

**Table 5 EMRFOGS performance and metrics measurements on the GSE33000 dataset with 39,279 features.** The bold and underlined entries in this table indicate the best value.

| Model | Measurements before/after | | | | AUC scores (after) | Selected features | $Global_{best}$ convergence iteration |
|---|---|---|---|---|---|---|---|
| | Accuracy | F1 | Recall | Precision | | | |
| NB | 0.71/0.84 | 0.72/0.84 | 0.71/0.84 | 0.74/0.84 | 0.93 | 70 | **41** |
| RF | 0.90/0.87 | 0.90/0.87 | 0.90/0.87 | 0.90/0.88 | 0.96 | **64** | 64 |
| SVM | 0.83/**0.90** | 0.83/**0.90** | 0.83/**0.91** | 0.84/**0.97** | 0.97 | 225 | 79 |
| XGBoost | **0.92**/0.89 | **0.92**/0.89 | **0.92**/0.89 | **0.92**/0.90 | **0.98** | 431 | 74 |

due to an increase in imbalanced data samples from 161 to 624 of GSE5281 and GSE33000, respectively. Moreover, it is conceivable that a high number of samples added noise and increased variability in complex relationships among data features, increasing the complexity for NB to capture the underlying representation of the data. Performance metrics for NB and SVM notably improved after EMRFOGS gene selection. Starting with NB, the results in Table 5 show a considerable 18.30% increase across all metrics, with an AUC of 0.93. The notable improvements highlight the EMRFOGS efficient gene selection of AD GSE33000. The SVM classifier performs the highest across all metrics with an improvement of approximately ~8.43% and ~15.47% in AUC scores. However, the performance of both RF and XGBoost classifiers declined by more than 3% following EMRFOGS gene selection. Given that RF primarily relies on features randomness to build multiple decision trees, and XGBoost gradient boosting model construct trees in a stepwise manner while attempting to reduce errors of previous step. Then, it possible to hypothesized that EMRFOGS removed certain features that had influenced both classifiers' superior performance before gene selection. With no further refinements of the $Global_{best}$, RF and XGBoost converged earlier to local minima at approximately iteration number 64 and 70, respectively (shown Figs. 10A and 10B). This impact of performance can be minimized through model's hyper-parameters tuning or hybrid feature selection strategy such as feature importance.

## GSE44768

The classification performance of SVM on the GSE447768 dataset, as shown in Table 6, outperformed all classifiers across all metrics. SVM's generalized performance across all the evaluated GSE datasets was the highest thus far. XGBoost efficient classification signifies a close correlation to the number of samples of GSE datasets, as the number of samples in GSE44768 is roughly half of GSE33000 dataset. Conversely, and as evident from previously evaluated GSE datasets, NB remained the worst-performing classifier before gene selection underperforming SVM by ~19.73%. This implies that the NB theoretical model's basic assumption—that every feature is statistically independent of every other feature—could be the reason behind its poor performance. However, NB classifier exhibited contrasting behavior as it significantly outperformed RF and SVM classifiers in all parameters with scores of 0.97% and 0.98% of AUC, see Table 6. This improvement was observed only on GSE44768 dataset which cloud imply that the selected set of genes by

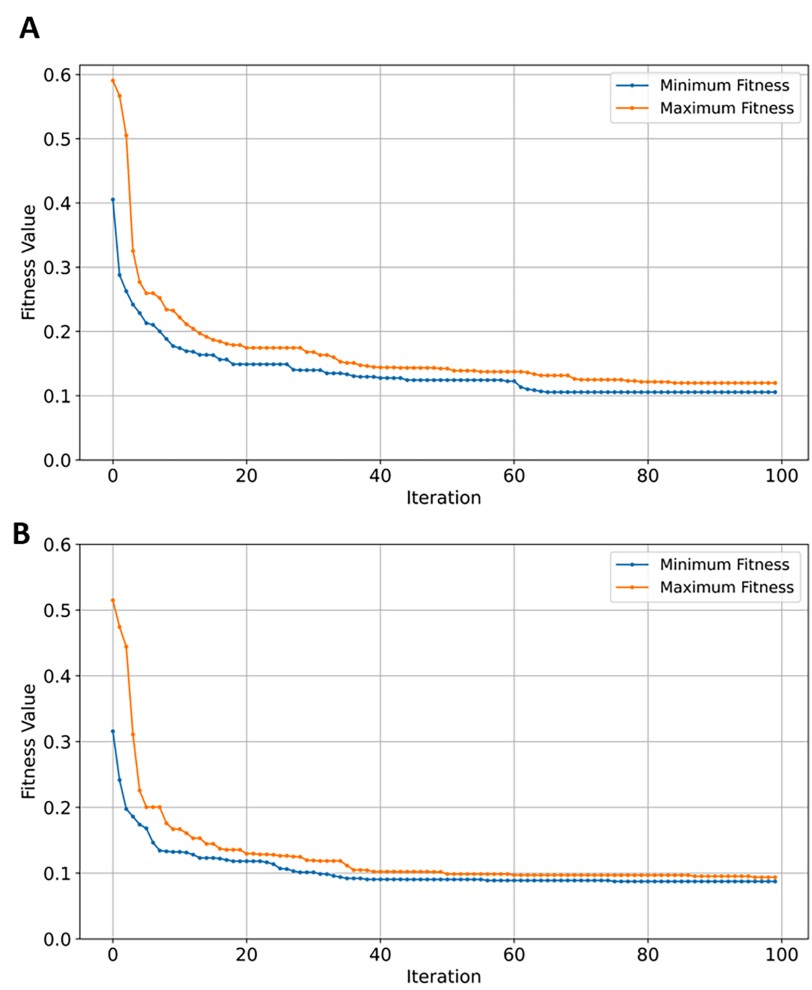

**Figure 10 Convergence of fitness minimum and maximum values of EMRFOGS for GSE33000 dataset.** (A) Random forest classifier, (B) XGBoost classifier.

**Table 6 EMRFOGS performance and metrics measurements on the GSE44768 dataset with 39,279 features.** The bold and underlined entries in this table indicate the best value.

| Model | Measurements before/after | | | | AUC scores (after) | Selected features | $Global_{best}$ convergence iteration |
|---|---|---|---|---|---|---|---|
| | Accuracy | F1 | Recall | Precision | | | |
| NB | 0.76/0.97 | 0.76/0.97 | 0.76/0.97 | 0.77/0.97 | 0.98 | 119 | **23** |
| RF | 0.84/0.84 | 0.84/0.84 | 0.84/0.84 | 0.84/0.84 | 0.90 | 134 | 38 |
| SVM | **0.91**/0.92 | **0.91**/0.92 | **0.91**/0.92 | **0.91**/0.92 | 0.98 | 168 | 65 |
| XGBoost | 0.88/**1.00** | 0.88/**1.00** | 0.88/**1.00** | 0.88/**1.00** | **1.00** | **59** | 83 |

EMRFOGS aligned with NB statistical model. Convergence behavior of EMRFOGS-NB illustrated in Fig. 11 is consistent—in comparison to Figs. 9 and 10—which further substantiates the previous analysis. The performance of SVM was marginally improved

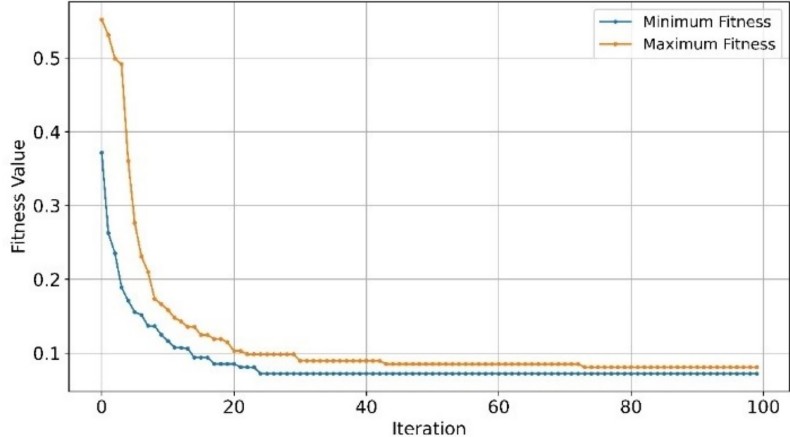

**Figure 11 Convergence of minimum and maximum fitness values of EMRFOGS for GSE44768 using NB classifier.**

marginally by 1%, in contrast to XGBoost perfect scores, which suggests overfitting behavior similar to GSE1297 dataset.

## GSE44770

SVM classifier performed the highest on GSE44770 dataset—388 samples and 39,279 genes—achieving accuracy scores of 0.89. The SVM classifier demonstrated consistency in class separation and error minimization in high-dimensional AD GSE datasets. In comparison to all previously tested GSE datasets, the classification performance and generalization of four learning models on GSE44770 is relatively similar. As shown in Table 7, a difference in efficiency of between the best and worst performing classifiers (SVM and NB) is measured at ~5.95%. Applying EMRFOGS gene selection significantly improved the performance of SVM by approximately 5.61% in accuracy, F1, and recall metrics, and by ~4.44% in precision. The decline in RF performance shows similar behavior to previously evaluated GSE33000 dataset. XGBoost, on the other hand, exhibited a significant improvement of ~5.81% with no potential of overfitting in contrast to GSE44768 and GSE1297.

Although, EMRFOGS gene selection improved NB performance by approximately 1.1%, NB remained the least effective classifier in all previously evaluated AD GSE datasets. The decline in performance of RF classifier on the GSE44770 dataset was similar to its performance on the GSE33000 despite retaining the lowest number of genes in both tests. This behavior can be attributed to its classification insensitivity, the effect of other objectives included in the optimization fitness function is diminished by the low number of selected genes. Thereby, forcing convergence to local minima in early stages of optimization and preventing the discovery of a new optimal solution with higher accuracy. This is evident in comparing the fitness convergence curves of EMRFOGS-SVM and EMRFOGS-RF in Figs. 12A and 12B, and to a greater extent, the evaluated datasets

**Table 7 EMRFOGS performance and metrics measurements on GSE44770 dataset with 39,279 features.** The bold and underlined entries in this table indicate the best value.

| Model | Measurements before/after | | | | AUC scores (after) | Selected features | $Global_{best}$ convergence iteration |
|---|---|---|---|---|---|---|---|
| | Accuracy | F1 | Recall | Precision | | | |
| NB | 0.84/0.85 | 0.84/0.85 | 0.84/0.85 | 0.84/0.87 | 0.93 | 314 | 97 |
| RF | 0.86/0.84 | 0.86/0.84 | 0.86/0.84 | 0.86/0.84 | 0.93 | **73** | 95 |
| SVM | **0.89/0.94** | **0.89/0.94** | **0.89/0.94** | **0.90/0.94** | **0.98** | 212 | 82 |
| XGBoost | 0.86/0.91 | 0.86/0.91 | 0.86/0.91 | 0.86/0.91 | 0.97 | 383 | **60** |

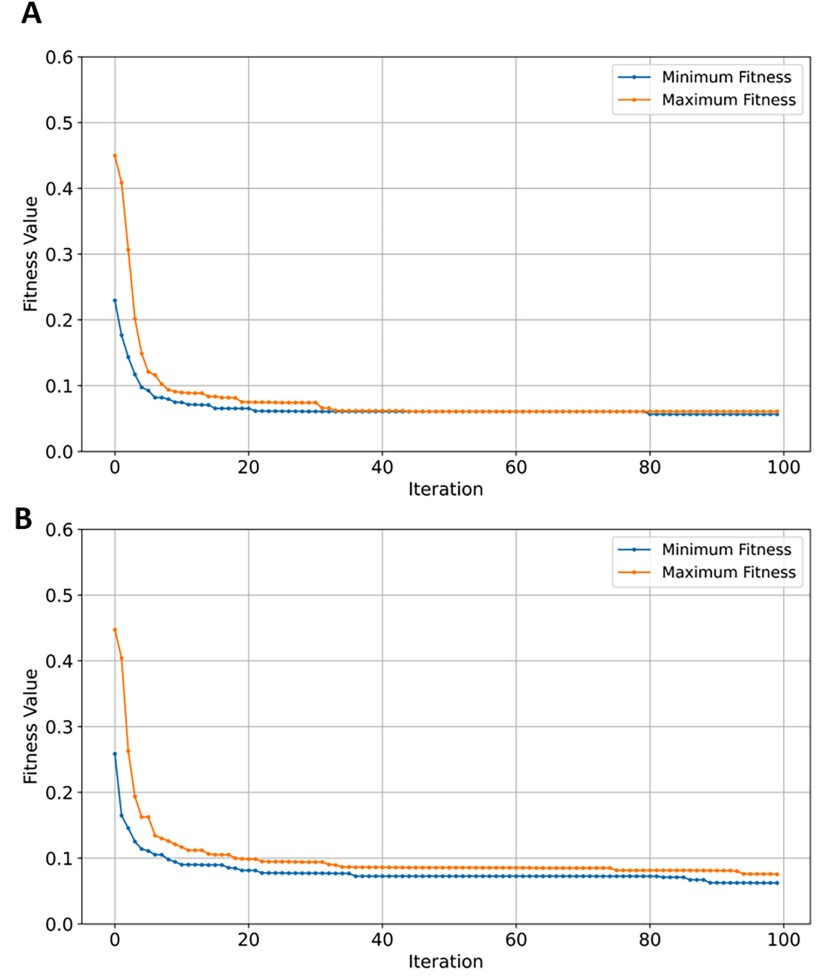

**Figure 12 Convergence of fitness minimum and maximum values of EMRFOGS for GSE44770 dataset.** (A) SVM classifier, (B) random forest classifier.

GSE3300, GSE44770, and GSE44768 have the same number of genes. This further substantiates the RF insensitivity failed to guide EMRFOGS exploration-exploitation to the optimal solution and slowed convergence to the $Global_{best}$ solution. Further exploration of different fitness functions and/or RF parameter tuning is recommended.

**Table 8** EMRFOGS performance and metrics measurements on the GSE132903 dataset with 42,178 features. The bold and underlined entries in this table indicate the best value.

| Model | Measurements before/after | | | | AUC scores (after) | Selected features | $Global_{best}$ convergence iteration |
|---|---|---|---|---|---|---|---|
| | Accuracy | F1 | Recall | Precision | | | |
| NB | 0.86/0.89 | 0.86/0.89 | 0.86/0.89 | 0.86/0.90 | 0.93 | 155 | **60** |
| RF | **0.88**/0.84 | **0.88**/0.84 | **0.88**/0.84 | **0.88**/0.85 | 0.92 | 151 | 78 |
| SVM | 0.77/**0.91** | 0.77/**0.91** | 0.77/**0.91** | 0.77/**0.91** | **0.94** | **46** | 99 |
| XGBoost | 0.84/0.84 | 0.84/0.84 | 0.84/0.84 | 0.85/0.84 | 0.93 | 154 | 88 |

### GSE1392903

The analysis of the GSE132903 dataset test results in Table 8 shows that RF attained the highest performance. The reduced number of samples seems to have influenced the performance of the RF classifier—despite the increase in number of genes from 39,279 in the three previously evaluated GSE datasets to 42,178. The low number of samples in GSE132903 contributed to RF classification efficiency. RF outperformed SVM by approximately 12.5%. In contrast, NB scored 0.86 across all performance metrics as the second-best classifier. SVM performed the worst with an accuracy of 0.77, while XGBoost classifier remained competitive with a score of 0.84 across the performance metrics.

After gene selection, EMRFOGS-SVM retained the best-performing classifier across all metrics with a performance improvement of ~18.18% and AUC score of 0.94. With no evident increase in performance, XGBoost performed the same, maintaining similar metrics even after gene selection. The number of data samples seems to significantly impact the performance of XGBoost, overfitting was observed in smaller datasets while larger GSE datasets necessitate a higher number of optimization iterations given that $Global_{best}$ converges at higher optimization iteration in comparison to other classifiers. NB classifier showed performance improvement of ~3.48% in addition to the solution fast convergence to only 155 genes at iteration 60, as opposed to RF with 151 selected genes and convergence iteration of 78.

SVM demonstrated a significant reduction in feature space, from 42,178 to just 46 (~99.8%). While NB, RF, and XGBoost maintained good balance between the convergence rate and the number of selected genes, the latter performed slightly better in terms of exploration as shown in Figs. 13A, 13B, and 13C, respectively. EMRFOGS-XGBoost was more active in exploring different regions of the complex search space during the first ~40 optimization iteration.

### Experiment results comparison with reviewed literatures

To further evaluate the gene selection performance of the proposed EMRFOGS method, Table 9 presents a comprehensive comparison of obtained results and those reported in the reviewed literature. This analysis seeks to contextualize our findings with the broader body of AD gene selection research. In Table 9, the letter 'M' in the dataset column indicates that the utilized GSE datasets in the study are merged together, while the column 'SG' represents the number of selected genes and term 'HR' indicates that only the highest

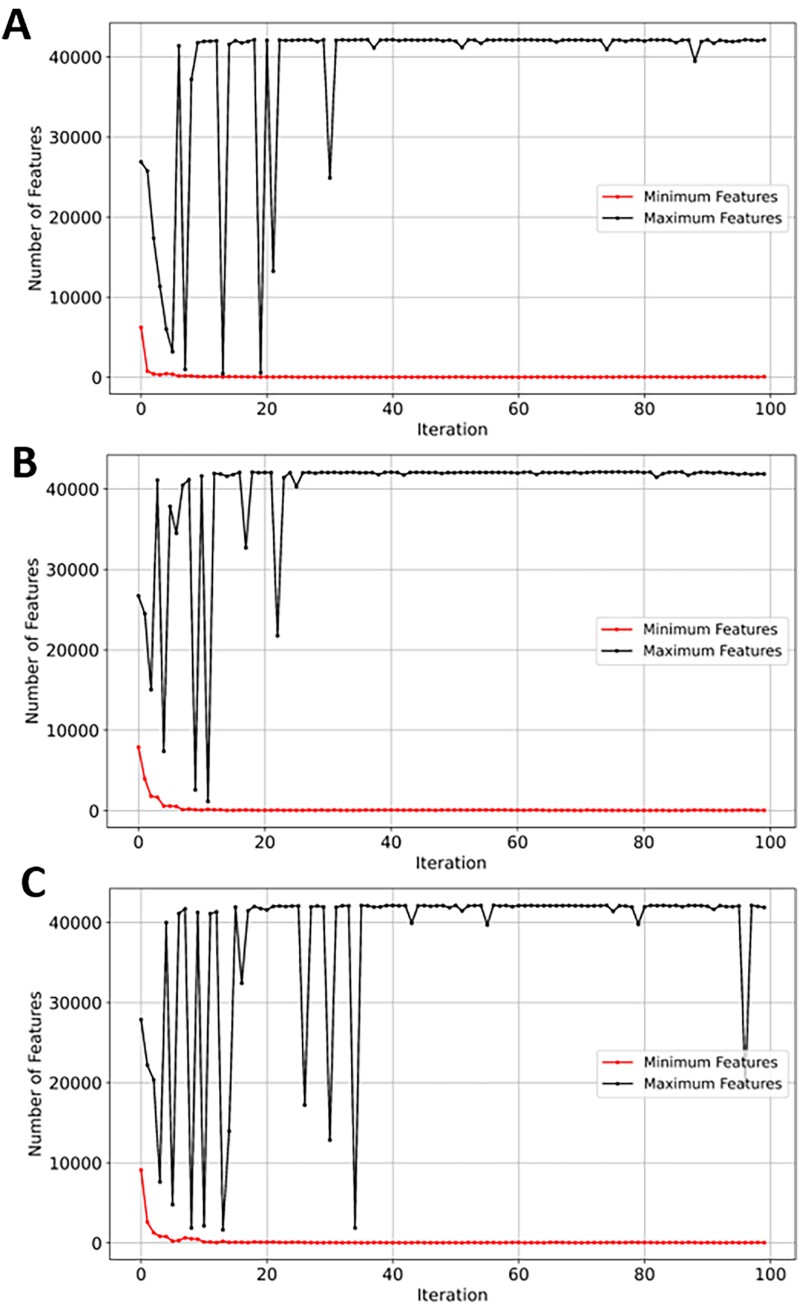

**Figure 13 Minimum and maximum number of features selected by EMRFOGS from GSE132903 dataset.** (A) Naive Bayes classifier, (B) Random Forest classifier, (C) XGBoost classifier.

ranked genes are considered after applying the gene selection method. Merging/integrating multi-omics datasets holds several key advantages such as increasing the prediction accuracy and improving the identification of molecular factors through techniques such as clustering, gene ontology, and protein-to-protein interaction.

**Table 9 Comparative analysis of EMRFOGS with reviewed methods.** The bold and underlined entries in this table indicate the best value.

| Ref./Year | Reviewed methods and their results | Dataset | SG | Scores | Proposed EMRFOGS Alg. | SG | Scores |
|---|---|---|---|---|---|---|---|
| *Zheng, Changgui & Huijie (2024)*/2024 | (Gene expression variance), Self-organizing map. | GSE1297 | 16 | ACC: 68 | RF | **4** | **ACC: 90** |
| *Finney et al. (2023)*/ 2023 | Hybrid (PCA, STRING), decision tree CRAT | M (GSE44770, GSE33000) | HR 1000 | – | RF | 73 | ACC: 93 |
| | | | | | RF | 64 | ACC: 96 |
| *Ahmed & Suhad (2023)*/2023 | Hybrid wrapper-based (Nomadic people optimizer (NPO), Information gain), SVM | GSE5281 | 1,115 | **ACC: 96** | NB | 162 | ACC: 92 |
| | | GSE132903 | 1,100 | ACC: 92 | SVM | **46** | **ACC: 94** |
| | | GSE33000 | 322 | ACC: 92 | RF | **64** | **ACC: 96** |
| *Wang et al. (2023)*/ 2023 | Hybrid (WGCNA, GO, and KEEG), LASSO | GSE132903 | 462 | **ACC: 94** | SVM | **46** | **ACC: 94** |
| *Zhang et al. (2022)*/ 2022 | Hybrid (WGCNA, PPI, STRING), LASSO-COX regression | GSE5281 | 959 | ACC: 90 | NB | 162 | **ACC: 92** |
| *El-Gawady, BenBella & Mohamed (2023)*/ 2022 | Hybrid filter-based (Chi square, ANOVA, Mutual information), SVM | M (GSE33000, GSE44770, GSE44768) | HR 1058 | **ACC: 97** | RF | **64** | ACC: 96 |
| | | | | | RF | **73** | ACC: 93 |
| | | | | | XGB | **59** | ACC: 1.0 |
| *Mahendran et al. (2021)*/2021 | Hybrid filter-wrapper (mRmR, Particle swarm algorithm (PSO), Autoencoder (AE)), Improved deep belief network (IDBN) | GSE5281 | **11** | **ACC: 96** | NB | 162 | ACC: 87 |
| *Chihyun, Jihwan & Sanghyun (2020)*/ 2020 | Hybrid (Differential gene expression, differential methylated position), deep neural network (DNN) | M (GSE33000, GSE44770) | **35** | ACC: 70 | RF | 64 | **ACC: 87** |

## CONCLUSION

Early diagnosis and treatment of Alzheimer's disease are regarded as critical steps in slowing its progression and improving patients' outcome. However, it remains a significant challenge primarily due to disease complex pathophysiological mechanism. The applications of data mining and machine learning algorithms in Alzheimer's microarray analysis had significantly improved the disease diagnosis accuracy, prognosis assessment, patient's tailored-treatment, and advanced our knowledge of the diseases at molecular level. However, processing high-dimensional microarray datasets using machine learning is surrounded by several challenges, such as overfitting and computational complexity. The curse of dimensionality, caused by the high-dimensional microarray datasets, is a major cause of several challenges facing the applications of machine learning in disease diagnosis. Hence, gene selection methods have been earmarked as a way of addressing such issue to increase AD prediction.

In this study, a wrapper-based method known as enhanced manta ray foraging optimization gene selection (EMRFOGS) algorithm was introduced to efficiently explore, identify gene relationships patterns, and reduce dimensional space of Alzheimer's gene expression datasets. Evaluations showed that the enhanced MRFO algorithm with sign random mutation, best rank, and hybrid boundary function significantly reduced the number of genes and improved prediction accuracy. Among the four evaluated classifiers, SVM consistently demonstrated high classification sensitivity and attained the highest classification metrics after the application of EMRFOGS gene selection method. This shows SVM strong adaptability to different complex and high-dimensional gene expression datasets. In contrast, the XGBoost classifier showed signs of overfitting in two datasets and marginal improvements observed in two others. Moreover, the number of selected genes was higher in comparison to other classifiers, despite achieving nearly similar iteration of convergence. This variability suggests that EMRFOGS-XGBoost requires extended optimization iterations (more than 100 iteration) and careful XGBoost hyperparameters tuning. Further still, naïve Bayes performance was also highly unstable and strongly dataset-dependent with signs of overfitting in one instance, which can be attributed to model's inherent assumption that features are statistically independent. Lastly, the overall performance of random forest appeared to be influenced by the number of samples in the dataset at hand. Nevertheless, RF remained competitive—performing comparably to or even better—to SVM and XGBoost in several instances, with two scenarios where the classifier attained the lowest number of selected genes.

Despite the efficient gene selection performance of the proposed EMRFOGS algorithm, three main limitations require further investigation. First, the study examined only a single transformation function (S1). In general, transformation functions can either restrict or enhance algorithm's ability to capture diverse feature relationships and patterns. Hence, exploring transformation strategies is essential to thoroughly assess the proposed algorithm's performance. Second, gene expression profiles vary significantly from one disease to another; therefore, it is challenging to determine whether the obtained efficiency and improvements in classification metrics are maintained for diseases beyond the context of Alzheimer disease. Therefore, evaluation of EMRFOGS on other diseases such breast or colon cancer is essential. Last, the evaluation in this study is limited to the dimensions of the evaluated gene expression datasets without scalability performance evaluation. Therefore, the proposed EMRFOGS algorithm may encounter significant performance degradation in real-world big data biomedical applications.

Last, several improvements are encouraged in future research, such as the integration of genes' biomedical knowledge and protein-to-protein interaction data as a preprocessing step or in the evaluation of the fitness function. Additionally, adaptive penalty function to eliminate weak solutions and hyperparameters tuning of classification models may aid in improving convergence speed and reduce the risk of overfitting.

### Funding
The authors received no funding for this work.

### Competing Interests
The authors declare that they have no competing interests.

### Author Contributions
- Zahraa Ahmed conceived and designed the experiments, performed the experiments, analyzed the data, performed the computation work, prepared figures and/or tables, authored or reviewed drafts of the article, and approved the final draft.
- Mesut Çevik conceived and designed the experiments, analyzed the data, authored or reviewed drafts of the article, and approved the final draft.

### Data Availability
The original data presented in the study are available at NCBI GEO: GSE5281, GSE33000, GSE1297, GSE44768, GSE44770, GSE132903.

Code is available in the Supplemental files.

### Supplemental Information
Supplemental information for this article can be found online at http://dx.doi.org/10.7717/peerj-cs.3064#supplemental-information.

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
