# Peer review of "Improving machine learning detection of Alzheimer disease using enhanced manta ray gene selection of Alzheimer gene expression datasets"

_PeerJ Computer Science, doi:10.7717/peerj-cs.3064_

## Round 0.1 · original submission · Major Revisions

The manuscript proposed an enhanced wrapper-based gene selection method (EMRFOGS) using machine learning to improve Alzheimer's disease detection from gene expression data. Reviewers highlighted several critical issues that must be addressed: clearly stating the study aim and research gap in the introduction, rigorously addressing potential overfitting through statistical validation (e.g., nested cross-validation or external datasets), providing biological interpretation of selected genes, performing formal statistical significance testing, and comparing the proposed method directly with standard MRFO and other recent methods. We would like to invite you to resubmit the manuscript after addressing these points that the reviewers raised will significantly strengthen the manuscript.

Reviewer 1 ·

Basic reporting

The manuscript used ML methods to increase the Alzheimer disease detection using gene expression datasets.

Experimental design

1. There should be an explicit study aim at the end of the introduction to clearly let readers know the purpose.
2. The authors should clearly tease out what has already been found and investigated in the introduction. On top of this, define the research gap as well.

Validity of the findings

Please add the strengths and limitations before the conclusion. Also, provide a clean conclusion at the end.

Additional comments

The quality of the figures and tables should be improved.

Reviewer 2 ·

Basic reporting

This manuscript presents a novel wrapper-based gene selection method termed EMRFOGS (Enhanced Manta Ray Foraging Optimization Gene Selection) to address the high-dimensional nature of gene expression datasets in Alzheimer’s Disease (AD) prediction. The authors improve upon the original Manta Ray Foraging Optimization (MRFO) algorithm by integrating three key enhancements, Sign Random Mutation (SRM), Best Rank (BR), and Hybrid Boundary Function (HBF), to improve the algorithm’s exploration and exploitation abilities. The enhanced approach is applied to six publicly available Alzheimer’s gene expression datasets, using four different machine learning classifiers (SVM, RF, XGBoost, and Naïve Bayes) to evaluate classification performance before and after gene selection. While the proposed method shows promise and is well-structured, the issues of overfitting, lack of biological validation, and statistical robustness need to be addressed before this work can be considered for publication.

Experimental design

1. In several cases, particularly with XGBoost and Naïve Bayes, the model achieves near-perfect accuracy and AUC scores after gene selection. While the authors mention possible overfitting, they do not rigorously address this with statistical safeguards such as nested cross-validation or external validation datasets. This undermines the generalizability of the results.
2. Although the method successfully reduces thousands of genes to a few dozen, there is minimal biological discussion or functional annotation of the selected genes. Without this, it is unclear whether the genes have clinical or diagnostic relevance beyond computational performance.

Validity of the findings

3. The performance improvements are reported descriptively, but the manuscript lacks formal significance testing (e.g., paired t-tests or Wilcoxon tests) to validate that EMRFOGS offers statistically meaningful improvements over baseline or other methods.
4. The results suggest that classifier performance is highly dataset-dependent. For instance, NB shows overfitting in one dataset but underperforms in others. A deeper analysis of variance or robustness across runs would be beneficial to understand algorithmic stability.
5. The use of a linear weighted sum for fitness evaluation (accuracy and feature count) lacks exploration into weight sensitivity or alternative formulations (e.g., multi-objective optimization), which could yield different trade-offs between accuracy and dimensionality.

Additional comments

6. While the manuscript includes a comparative table of past literature, it would benefit from direct experimental comparisons (e.g., rerunning existing methods on the same datasets), especially against more recent deep learning-based gene selection frameworks.

Annotated reviews are not available for download in order to protect the identity of reviewers who chose to remain anonymous.

Reviewer 3 ·

Basic reporting

Ahmed & Çevik proposed using an enhanced framework of Manta Ray Foraging Optimizer (MRFO), a wrapper-based method, to improve the gene selection in gene expression data and their prediction accuracy for Alzheimer’s disease sample status. The authors used Best Rank to enhance exploration of best subset in feature space and Sign Random Mutation to enhance exploitation. The authors benchmarked their wrapper approach on six GSE Alzheimer’s disease datasets based on different classifier such as SVM and random forest. Compared to previously identified genes and prediction performance, the authors found their method identified fewer genes and maintained a very high AUC. The manuscript is written well, and results presented clearly. With that said I have several comments.

Experimental design

1. Can the authors add a comparison with standard MRFO approach in the result section?
2. I would be curious to see the proposed method performance in a new test dataset, if the authors can find a separate GSE data to make prediction.

Validity of the findings

1. There is typo in line 132. The AUC should be 0.94, no percentage sign. Same type occur in line 181.
2. Please define EMRFOGS in line 545.

---

## Round 0.2 · accepted · Accept

Thank you for responding to the comments. We don’t have further suggestions. The manuscript will be transferred to the journal staff and production team for publication. Please feel free to contact us if you have any questions in the later stages.

Reviewer 3 ·

Basic reporting

N/A

Experimental design

N/A

Validity of the findings

N/A